# Basic school pupils' food purchases during mid-morning break in urban Ghanaian schools

Deda Ogum-Alangea[1]*, Richmond N. O. Aryeetey[1], Heewon L. Gray[2], Amos K. Laar[1], Richard M. K. Adanu[1]

**1** Department of Population, Family and Reproductive Health, University of Ghana School of Public Health, Legon, Ghana, **2** College of Public Health, University of South Florida, Tampa, Florida, United States of America

* dogumalangea@ug.edu.gh

**Data Availability Statement:** All relevant data are within the paper and its Supporting Information files.

## Abstract

### Background

Unhealthy food vending can expose children to malnutrition and other diet related challenges such as obesity. This study sought to describe types and sources of food in basic schools in urban Accra, and to describe food purchases by pupils.

### Methods

This was a cross-sectional study of five basic schools (3 public; 2 private) and 644 pupils in the Ga-East Municipality in Ghana. Check-lists were used to document available sources of foods during school hours. Pupils were intercepted after making purchases during breaktime and the type, cost and sources of foods purchased documented. Energy content of foods were read from labels when available or estimated using the Ghana Food Composition database when unlabelled. Frequencies and crosstabs were used to compare food type by source and school type.

### Results

Foods were purchased from school canteen, school store, private stores, and 'table-top' vendors. Meals were most frequently purchased (38%) although single purchases were sweetened drinks, savoury snacks and confectioneries. About 53% of retailers located within the schools sold relatively healthier food options. Similar foods with comparable energy content were purchased within and outside of school.

### Conclusions

Basic schools in urban Ghana provide ready access to energy dense food options, which are purchased by pupils both within and outside of school premises. Timely interventions inclusive of school food policies can encourage healthier diets among pupils.

**Funding:** DOA and RNO received support from the "Building Stronger Universities in developing countries (BSU)-PHH grant through the college of Health Sciences, University of Ghana, Legon for the collection of data used in the current study. The funders had no role in study design, data collection and analysis, decision to publish, or preparation of the manuscript.

**Competing interests:** The authors have declared that no competing interest exist.

**Abbreviations:** GHS, Ghana Cedi; JHS, Junior high school; MMDA, Metropolitan, Municipal and District Assemblies; mRFEI, modified Retail Food Environment Index.

## Introduction

The availability, sources, purchase, and consumption of foods within the school setting constitutes the school food environment. The school food environment, is important because it is a known driver of dietary behaviour among children and adolescents in school [1]. Providing healthy food options at school is linked with learning of appropriate dietary preferences as well as capacity to overcome barriers to sub-optimal dietary practices [2]. Conversely, vending unhealthy food can expose children to malnutrition and other diet-related non-communicable diseases (NCDs). In Ghana, there is limited evidence on the school food environment and pupil purchasing behaviour [3, 4]. More scarce is evidence on pupil food purchases at school in the presence or absence of school provided meals or regulation.

Evidence from developed country settings on actions to regulate the availability of unhealthy food options to pupils at school shows mixed benefits. Prohibiting energy-dense foods in the school premises have increased the consumption of more healthful options provided in the school [5–7]. However, this benefit may be undermined by pupils gradually opting out of school meals and rather bringing less healthy packed lunches or prohibited food to school [2, 8, 9]. There have also been reports that some entrepreneurial pupils purchase these prohibited foods in bulk out of school and resell to their colleagues at school [10]. A comparison of school lunch and packed lunches, following the regulation of compulsory food standards showed that school lunches had more favourable caloric and nutrient profiles compared to packed lunches [11, 12]. Irrespective of the challenges with regulation, policies related to packed lunches or school lunches have been suggested among efforts at achieving healthy dietary intakes of pupils at school [11].

Schools face structural and resource constraints in providing adequate canteen services. Studies examining school food and pupil patronage revealed that pupils refuse to patronize school food for several reasons. These include long queues, limited sitting places in canteen, insufficient food variety, the limited choice of meal components, portion sizes, quarrels with colleagues and catering staff, as well as non-inclusion of pupils in school meal planning [9, 13, 14]. Adolescents especially, need the social environment during break periods to build relationships and exercise autonomy [15]. Thus, any perceived 'waste' of their time or lack of respect, as usually happens in school diners broods dissatisfaction [14]. There is evidence that although pupils are aware of the healthiness of school food, they will rather go out to purchase packed food due to some of the unaddressed social challenges involved in getting school food [16].

Schools have been identified as ideal sites for interventions to improve child and adolescent diets and health in general [17]. School-based policies and programmes that support healthy eating and physical activity are crucial in reversing the childhood obesity epidemic as well as other diet-related challenges [17]. Policies and interventions targeting healthy eating in school settings include provision of healthier options through school breakfast and lunch, increasing availability of fruits, reducing availability of energy-dense snacks in school, preventing pupils from accessing food outside school premises, guidelines on content of foods and food retailing in schools [18, 19].

In Ghana, there are no specific policies on food sold in the school environment. The Public Health Act of 2012 (Act 851) is the main regulatory instrument, which requires that food offered to consumers be wholesome. It also requires registration for businesses supplying food for sale as well as food hygiene certification for food vending staff. Metropolitan, Municipal and District Assemblies (MMDA) are expected to enforce vendor certification, and hygiene standards. However, the laws do not provide specific provisions for location and nutritional content of foods offered to consumers. Heads of schools or parent-teacher associations have

power to regulate the school food environment regarding what is allowed for sale. Meanwhile, heads of schools often lack the technical expertise in addressing the situation [20, 21].

In selected Ghanaian Public basic schools, pupils in kindergarten and primary classes one through six may be provided with one hot meal if they are participating in the School Feeding Programme (SFP). However, some private schools may have a school lunch programme and may allow selected food vendors into the school to sell a variety of foods, run shops that sell snacks or insist on packed lunches. Most of the public schools and a few private schools are not fenced and may allow pupils to make food purchases outside the school premises.

Creating a healthy food environment for pupils at school will require coordinated efforts to control unhealthy food exposures outside the school compound since children are also exposed to food vendors along their journey to/from school [22]. An expert assessment of government action on implementation of healthy food environment policies in Ghana concluded that implementation was insufficient [23]. There is also reported concentration of unhealthy foods and beverages in formal establishments in urban Ghana [24] with high evidence of the dominance of unhealthy foods in commercial advertisements in higher education [25]. Literature on how pupils interact with the prevailing food retail environment at schools in Ghana is limited [3, 4].

The primary goal of this paper is to describe the food sources, types and nutrient contents of foods available to pupils and typical purchases at school during mid-morning breaktime, Evidence from this study will contribute to the body of knowledge on food consumption at school and inform policies targeted at improving healthy eating among basic school pupils in urban Ghana.

## Methods

### Study design and participants

This current study was implemented in urban and peri-urban settings in the Ga East Municipality of Ghana. This cross-sectional study involved upper primary (grade 4–6) and Junior high school (JHS) pupils enrolled in 5 selected basic schools (3 public; 2 private). These five schools are a sub-sample of 24 schools that were selected for inclusion in a study assessing the determinants of obesity among basic school pupils. Purposive selection of the five schools was based on typical food environments identified among the 24 schools. The characteristics of the five schools are as follows: A—Fenced and gated private school and pupils are not allowed to go outside of school during break time; B—Fenced and gated private school but pupils can go out during break time; C–Fenced and gated public school that is part of a cluster of schools; D—Unfenced public school with no clear demarcation of school premises and having food vendors easily accessible to pupils; E—Unfenced public school with clearly demarcated school premise. A total of 644 pupils were randomly intercepted during break time after making food purchases at school in April 2013.

### Ethics approval and consent to participate

This study was a constituent of a larger study (study on determinants of childhood obesity in basic schools) for which ethical approval obtained from the Noguchi Memorial Institute for Medical Research (reference number NMIMR-IRB CPN 102/11-12). Parental consent was obtained for eligible pupils in selected schools. Additionally, verbal assent was obtained from participating pupils before interviews were conducted. All field tools were assigned unique Identification numbers which were used on forms and during data entry. Thus, responses could not be linked to participating individuals in the study.

## Study materials

Separate checklists were used to record foods available within the school premises and also within a 100-meter radius around each school. The checklists were designed to capture information on typical food sources within schools and those outside of the schools. The checklist used within school premises documented food items available in sources such as snack bars, canteens, other stores in premises, 'table top vendors'(items are displayed on top of a table either mobile or fixed to a venue), hawkers, and beverage vendors.

The checklist used to document foods and sources outside of the school premises captured food items available in stores, homes, kiosks, "table tops" and beverage vendors. Type of food items on sale by vendors both within and outside the school premises were classified as: 'meals'; 'packaged snacks/ biscuits'; 'local snacks (roasted plantain/corn/nuts etc.)'; 'fried foods'; 'pastries'; 'sweetened/non-fizzy drink'; 'fizzy drinks'; 'confectioneries/sweets (toffees/ candies/lollipops/ice cream) '; 'kebabs'; 'fruits'; or 'fruit juices'.

A structured questionnaire (S1 File) was used to document foods typically purchased by pupils during break time at school. The questionnaire documented the school type, sex and age of child, list of food items purchased, type of food item(s), the amount (g) and unit cost of purchase(s).

## Study procedures

In order to understand the typical food environment during break time at school, the research team visited each school on a minimum of three different occasions to both observe pupil purchasing behaviour during break time and also to list the available vendor types. Within one week prior to the interception study, research assistants listed all vendor types as well as the foods available on sale. The checklist was used to document both the number of different vendor types as well as the different food types available on sale.

The research team also took samples of food items in typical portions purchased by pupils and weighed. A database was compiled containing a list of different varieties of food types, weights of typical portions, unit costs as well as detailed comments on differences (cost and weight). For local foods that are sold based on customer indication, the research team purchased typical combinations of foods as observed earlier in schools and assessed the weights of individual constituents in composite foods. For example, pupils typically purchased *wakye* (cooked rice and beans) in the following manner: GHS 0.50 *wakye* + GHS 0.20 of (*gari*+ spaghetti). We then documented the actual weight of GHS 0.50 worth of *wakye* from at least three sources and used the mean weight for pupils who reported buying equivalent quantities. Other accompaniments like boiled egg, beef, chicken, or '*wele*' (cooked cow hide) were also weighted separately and added if purchase was reported. Database compilation was necessary because pupils who went outside of the school premises to buy meals are more likely to have consumed foods at the point of purchase compared to if they purchased snack items. At the end of each day's work, newly identified foods were procured, weighted, and included in the existing database.

On the day of interception, research assistants positioned themselves at vantage locations between the playground (where available), vendor locations, and classroom block, just before start of the mid-morning break period. Pupils returning from the vendor locations towards playground or classroom block were intercepted and asked if they have made any purchase(s) during the period. Items were inspected by research assistants (if present) and weighed with a simple food scale (Ohaus compact scale). Items reported as purchased by pupil but not available for weighing (consumed) were listed with its description, unit cost, and place of purchase. Subsequently, information on the weight of the food item was collected/updated from the

database of foods and drinks obtained from vendors in and around the school. Duplicate interviewing of pupils was minimized by asking the pupil if they had previously been intercepted.

The school food retail environment was benchmarked using an adaptation of the modified Retail Food Environment Index (mRFEI) score [26], which is a proportion of more healthful food retailers in each school. Vendors offering complete meals, fruits, local snacks, milk and cocoa products were grouped under healthier options and those offering other food types classified as less-healthier options.

### Data analysis

Data were entered into Microsoft Access and analysed in SPSS version 16. Frequencies were used to summarize food sources and food types available to pupils. The types of foods purchased by children were defined as multiple response data sets using SPSS. Frequencies were also used to report food types frequently purchased. The mRFEI score was calculated as: number of healthier food retailers divided by the total number of food retailers in the school.

Caloric content of packaged snacks was obtained from labels of packaged foods. Foods without nutrition content information were weighed and analysed using the Ghana food composition database [27] incorporated into ESHA FPRO version 10.0.0.

### Results

Mean [SD] age of respondents was 12.8[1.8] years, ranging from ages 9 to 17 years. The majority (63%) of respondents were girls; two-thirds were enrolled in public schools. A total of 1,124 foods and drinks were purchased by the 644 pupils intercepted during the study. About 78% of all breaktime purchases were made from food vendors within the school compound.

Hawkers in school compound (ie all the public schools) mostly offered pastries, fried foods, and confectioneries; only one hawker sold a ready-to-eat meal to pupils. No hawkers were observed in the private schools included in the study. The public schools studied did not have canteens and snack bars but were present in the private schools. Vendors who sold their food items on "table tops" were present in all public schools (range between 8 and 17 vendors) and in one private school (2 vendors). The 'table-top vendors were selling a variety food types with the exception of fruit juices. A branded beverage vendor was observed in one public school selling fizzy drinks, fried foods, pastries, packaged snacks, fried sausages, and eggs.

Seventy-seven food vendors were observed within the 100-meter radius around the five schools. However, disproportionately more vendors were situated around public schools compared to private schools (65 vs 12). Figs 1 and 2 show the availability of foods based on number of vendors selling items and the type of schools.

Multiple response analysis was conducted to examine frequent purchases since majority of pupils made more than one food purchase. The frequency of purchase of various types of food by pupils across public and private school and source of the food are shown in Tables 1 and 2 respectively. Energy-dense foods (pastries, biscuits, fried foods and sugar-sweetened beverages) formed 46.6% of all purchases; 50.3% of all purchases were made inside schools while 44.8% was purchased off school premises.

An assessment of the modified food retail environment of the five schools showed that schools generally recorded worsening retail environment with greater number of food vendors within the school compound. The two private schools had higher mRFEI scores (50% and 100%) compared to the Public schools (26%, 40% and 48%). Energy content of foods purchased by pupils are shown in table in S1 Table.

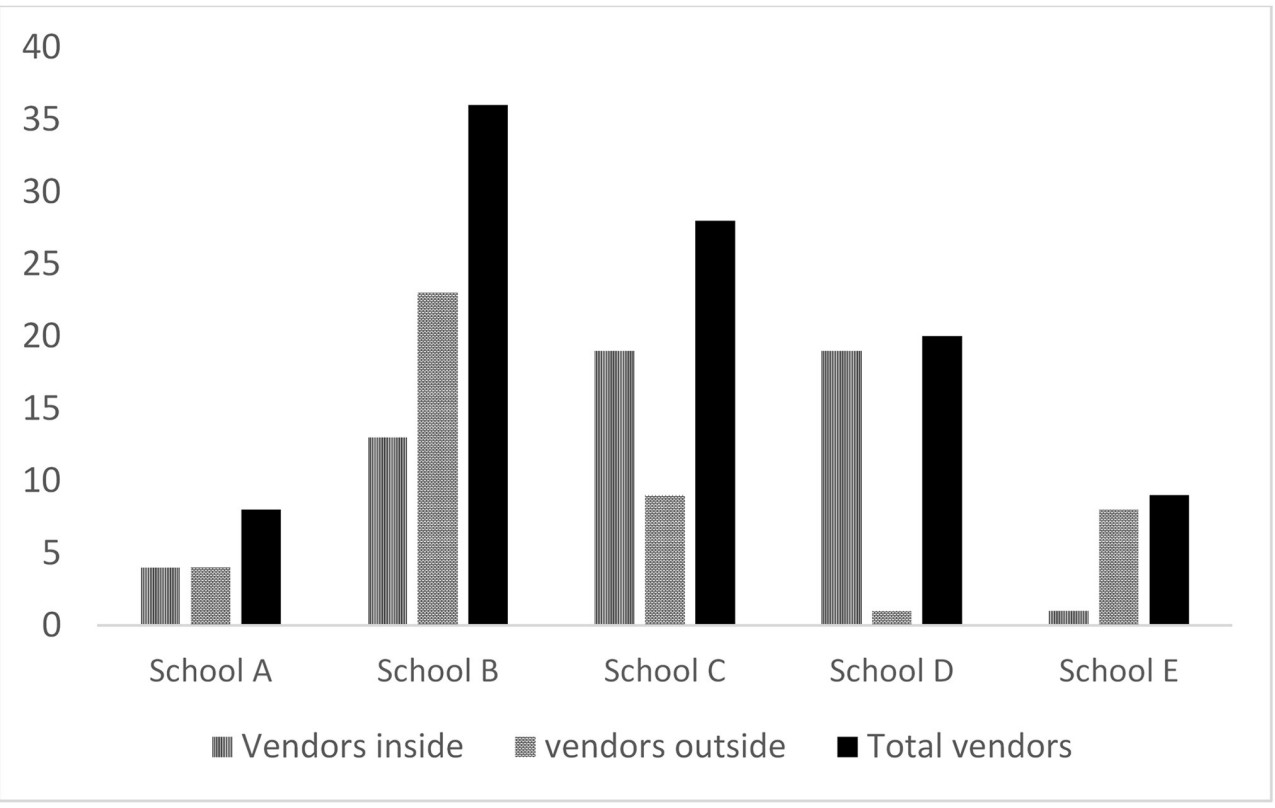

**Fig 1. Number of vendors offering foods and drinks to pupils.**

## Discussion

This study has documented the type of foods and drinks frequently purchased during mid-morning breaktime and the available sources of foods in selected schools in urban Accra. Evidence from our study suggests that pupils have access to a wide range of healthy and unhealthy food options from different vendors located within and around the school. The most frequent purchase made was 'meals'; although single food purchases were primarily sweetened drinks, confectioneries or pastries.

Overall proportion of healthier food retail points in and around schools visited was 53.4%, an indication that slightly more retailers sold healthy food in schools compared to less healthy ones. However, the proportion of healthy retail points reduced as the overall number of retail points increased resulting in lower scores reported in public schools (Private = 76.5% vs Public = 38%). It is also worthy of note that although healthier food options were present within the school compound, pupils were highly exposed to energy-dense options due to the wide variety of unhealthy options accessible to the pupils. This has daily and long-term implication for pupils' food choices at school since the presence of unhealthy options increases risk of purchase and consumption [28]. There is evidence that children consume less of healthy foods (fruits, vegetables, milk) when they have access to energy-dense foods [29, 30]. Beyond the influence of appearance [3] and peer choices [31], food choice and purchase by pupils is a complicated process that goes beyond satisfaction of hunger [14]. Social relationships can be formed and maintained around food and eating [32] and this is especially critical during adolescence [15, 33].

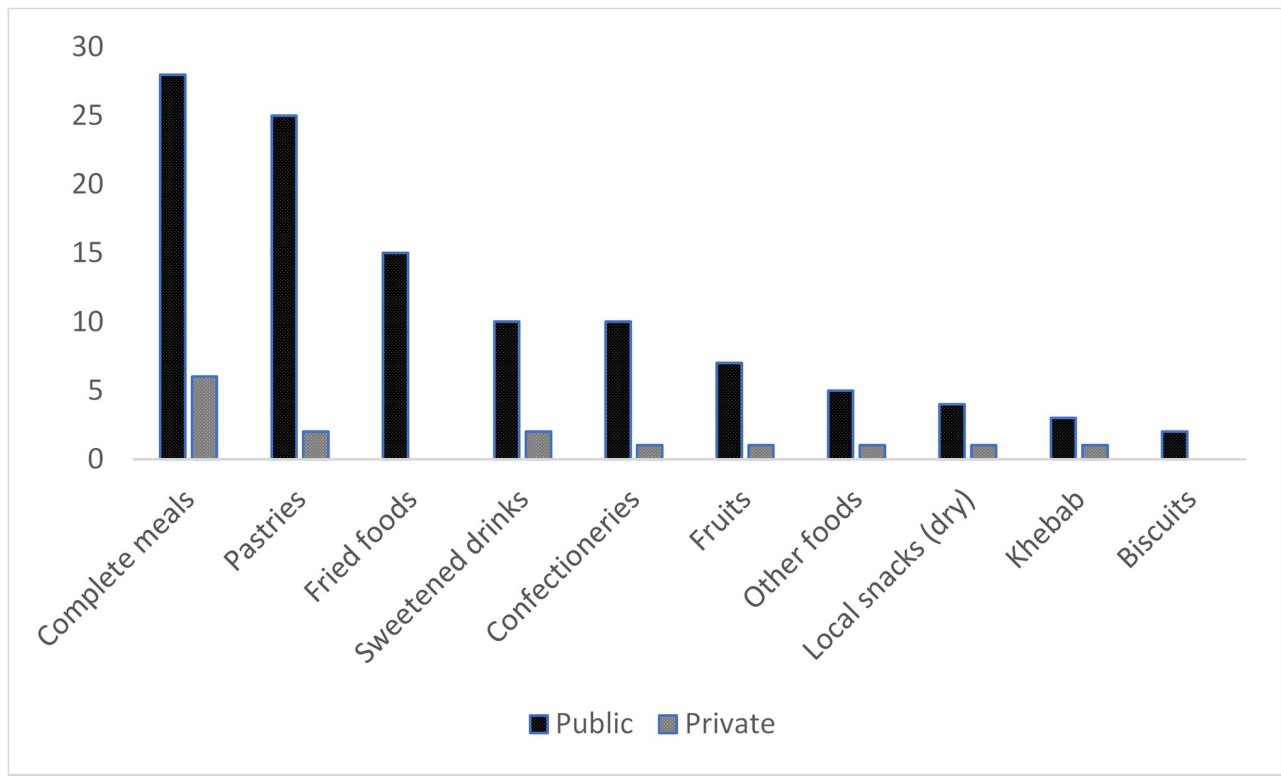

**Fig 2. Number of vendors offering specific food types within school premises.**

The building and nurture of unhealthy eating behaviours among pupils is a public health concern especially when unhealthy dietary habits are known to worsen with age [34]. Efforts to improve consumption of healthier foods as well as the food environment in schools can be done through nutrition education, policy, structural and environmental changes [2, 17]. Although evidence of long-term benefits for child health due to school food and food policies is lacking, the opportunity to influence dietary habits of a large population over a decade is

**Table 1. Frequency of pupil purchase of foods during mid-morning break period by school type.**

| Food Type | Frequency of purchase (percentage) | | |
|---|---|---|---|
| | Public school (n = 830) | Private schools (n = 294) | Total (n = 1124) |
| Complete meals | 343(41.3) | 82 (27.9) | 425 (37.8) |
| Sweetened drinks | 154(18.6) | 20 (6.8) | 174 (15.5) |
| Pastries | 74 (8.9) | 61 (20.7) | 135 (12.0) |
| Confectioneries | 44 (5.3) | 8 (2.7) | 52 (4.6) |
| Fried foods | 32 (3.9) | 30 (10.2) | 62 (5.5) |
| Milk & Cocoa drinks | 37 (4.5) | 19 (6.5) | 56 (5.0) |
| Fruits | 26 (3.1) | 22 (7.5) | 48 (4.3) |
| Local snacks (dry)[1] | 23 (2.8) | 13 (4.4) | 36 (3.2) |
| Biscuits | 27 (3.3) | 7 (2.4) | 34 (3.0) |
| Others[2] | 65 (7.8) | 25 (8.5) | 90 (8.0) |

[1]. Includes roasted plantain, corn, and yams, *'darkwa or zowe'*.

[2]. Other foods include fried sausages, fried eggs, boiled eggs, soybean khebab, fried chicken, which were bought by pupils and consumed alone or with drinks.

**Table 2. Frequency of pupil purchases during mid-morning break period by source of foods.**

| Food Type | Frequency of purchase (percentage) | | |
|---|---|---|---|
| | In school (n = 871) | Out of school (n = 253) | Total (n = 1124) |
| Complete meals | 325 (37.3) | 100 (39.5) | 425 (37.8) |
| Sweetened drinks | 148(85.1) | 26 (14.9) | 174 (15.5) |
| Pastries | 100 (73.3) | 35 (25.9) | 135 (12.0) |
| Confectioneries | 41 (78.8) | 11 (21.2)act | 52 (4.6) |
| Fried foods | 41 (66.1) | 21 (33.9) | 62 (5.5) |
| Milk & Cocoa drinks | 49 (87.5) | 7 (17.9) | 56 (5.0) |
| Fruits | 32 (66.7) | 16 (33.3) | 48 (4.3) |
| Local snacks (dry)[1] | 26 (72.2) | 10 (27.8) | 36 (3.2) |
| Biscuits | 29(85.3) | 5 (14.7) | 34 (3.0) |
| Others[2] | 75 (83.3) | 15 (16.7) | 90 (8.0) |

[1]. Includes roasted plantain, corn, and yams, *'darkwa or zowe'*(snack made from maize meal, peanuts and spices).

[2]. Other foods include fried sausages, fried eggs, boiled eggs, soybean khebab, fried chicken, which were bought by pupils and consumed alone or with drinks.

limitless [35]. Thus, food vending in and around basic schools in Ghana need to be regulated beyond health certification of vendors [3]. Even though the public schools involved in the current analysis were covered by the school feeding programme, only 8% reported the SFP as their main source of food during school hours in the bigger study [36]. Additional efforts should be directed towards providing nutrition education and making safe and nutritious foods more available to pupils to foster healthy dietary practices.

Mid-morning breaktime (9:30am to 10:30 am) presents an opportunity for pupils to either eat a meal if they skipped breakfast or get a snack. Pupils are more likely to meet their daily energy and nutrient requirements if they ate meals compared to energy-dense snacks or sweets. However, for pupils in private schools, who are more likely to be offered a compulsory lunch, the tendency to avoid heavy meals during mid-morning breaktime is higher compared to pupils who are relatively free to choose the type of food to purchase at lunch or other times. This assertion is supported by our data (see Table 1).

The patronage of complete meals in this setting is not without hygiene and quality concerns [37]. Yet, the absence of school-managed canteens offering cooked food outside lunchtime (where available) leaves pupils without any option than patronizing foods sold by vendors within school and outside the school. We observed in this study that pupils had to purchase sachet water after buying complete meals from vendors since the provision of clean drinking water with meals was absent. This situation could promote the replacement of water with sugar-sweetened drinks which were sold for a similar price. The danger here is that children generally may consume less than their fluid requirements and be at risk of voluntary dehydration [25]. Proper hydration of children is known to have positive impacts on mood and cognition [38], and schools must provide pupils with free clean drinking water. This is also to ensure that all pupils can be adequately hydrated when in school.

Sugar-sweetened beverages and pastries were the second most frequently purchased food during breaktime and this is similar to pupil purchases reported elsewhere [8, 12]. Evidence from an earlier study among pupils in the study area revealed that purchases were greatly influenced by radio and TV advertisements [3]. Our inventory of foods sold in and around the schools showed that fizzy drinks were not readily available for sale to pupils as is the case for schools elsewhere [39]. The unpopularity of carbonated drink sales in basic schools is probably

because of its relatively higher cost compared to the other sweetened drinks and vendors perceived lower patronage from pupils.

Similar to the findings of an earlier study among pupils in Ghana [3], patronage of confectioneries and savoury snacks was low and did not support the expected higher consumption of these foods due to the absence of regulations in the schools studied. It is possible that pupils who were just not hungry preferred drinks to confectioneries or savoury snacks. Culturally also, there is belief that eating 'sweets' in the morning causes 'stomach disturbances' and worm infestation. Collecting data on purchase throughout the day may be needed to test this hypothesis to explain consumption behaviours related to confectioneries and savoury snacks.

Some high-protein foods (e.g. fried sausages, eggs, soybean kebabs and fried chicken) purchased by pupils have the potential to improve the dietary protein intakes of pupils. Although street food may contribute to consumption of animal source foods and overall improvement in diet quality [40], concerns regarding fat content due to cooking methods may present challenges in the control of fat intakes among the population.

Earlier studies among Ghanaian adults [41] and children [42, 43] have reported low fruit consumption, which may explain low pupil fruit purchases in this study. Amfo-Ayeh [3], reported that only 9.3% of pupils preferred to buy fruits compared to 45.6% who preferred to buy energy-dense foods during break time. The evidence suggests that pupils may have modelled adult fruit purchasing and consumption behaviour from the general population.

Children are also known to be price-sensitive consumers [9]; and will be less willing to buy fruits due to relatively higher prices. Based on food price data collected in this study, fruits were only less priced compared to meals. Pupils will thus find purchasing of other high-calorie foods and drinks more as getting best value for their money. Comparing food prices in current study to a previous study [3] shows that fruit prices have risen over a period of two years whiles prices of sweetened drinks, pastries and sweets have reduced relative to cost of complete meals.

Fried foods made 5.5% of purchases. The frequent consumption of fried foods has the potential to negatively impact BMI due to elevated caloric consumptions associated with fried food fats [44] or through gene-fat interactions among persons genetically susceptible to obesity [45].

Our study found that school-managed shops mainly sold packaged drinks, pastries, and savoury snacks. These high-calorie foods present an opportunity for excessive calorie consumption, especially in private schools where canteen- provided- lunch was virtually compulsory to pupils. While schools will readily agree to making school food environments healthier, they may not be motivated to reduce the availability of these foods due the significant profits made on these highly patronised goods [46].

The secondary goal of this study was to describe the energy content of foods and drinks frequently purchased by pupils. We found that foods purchased inside the school premises were not different from those purchased outside of school. Thus, comparison could not be made on the energy content of foods purchased based on source.

A strength of this study is its ability to provide empirical evidence on what pupils purchase when at school in typical basic school food environments in urban Ghana. This is particularly useful in shaping policy amid the insufficient government regulation of food vending and environments in general. Currently, evidence is lacking on how pupils interact with their food environment when at school especially in LMICs including Ghana. This study has demonstrated that effective interventions for improving diets among basic school pupils will need to target food environments beyond the school fence in addition to comprehensive school food policies.

A limitation of this study is the non-inclusion of pupils in lower primary classes (from Kindergarten to year 3). Wills et al [13] reported different food purchasing and consumption

behaviours among younger and older pupils at school. The application of our findings, thus, may be limited to basic school pupils in upper primary classes and Junior High school. Another limitation is the possibility that changes in the food environment, due to the ongoing nutrition transition, may have occurred between the time of data collection and publication of this article. The authors believe that the effect of these changes is more likely to have impacted on the available options rather than the broad categorisation of food types as reported in this study. An information we believe is very relevant to inform content of policies to control unhealthy food exposure to basic school pupils.

## Conclusions

Basic school pupils in urban Ghana have ready access to energy dense food options both within and outside of school premises. The most frequent purchases made by pupils during mid-morning breaktime were complete meals which were obtained from vendors within and outside of school. There was no difference between foods available to pupils inside and outside of schools regarding food type or energy content.

Food environments in Ghanaian basic schools need to be regulated to encourage healthy eating habits among pupils. Pupils will greatly benefit from healthy food environments as part of global efforts at controlling unhealthy diets, obesity and other nutrition-related NCDs. Promulgating a school food policy that clearly describes the approved content of foods to be made available to pupils at school will greatly improve school food environments in basic schools.

## Supporting information

**S1 File. Questionnaire.**
(DOCX)

**S1 Table. Energy density of foods frequently purchased by pupils at school.**
(DOCX)

**S1 Data.**
(CSV)

## Acknowledgments

The authors gratefully acknowledge support received by the first author from the Building A New Generation of Academics in Africa (BANGA-Africa) project, University of Ghana to write the manuscript. We also acknowledge the contribution of all Research assistants, participating children, parents and head-teachers without whom this data would not be available.

## Author Contributions

**Conceptualization:** Deda Ogum-Alangea, Richmond N. O. Aryeetey, Amos K. Laar, Richard M. K. Adanu.

**Data curation:** Deda Ogum-Alangea, Richmond N. O. Aryeetey, Heewon L. Gray, Amos K. Laar, Richard M. K. Adanu.

**Formal analysis:** Deda Ogum-Alangea, Heewon L. Gray.

**Funding acquisition:** Deda Ogum-Alangea, Richmond N. O. Aryeetey, Richard M. K. Adanu.

**Investigation:** Deda Ogum-Alangea, Amos K. Laar, Richard M. K. Adanu.

**Methodology:** Deda Ogum-Alangea, Richmond N. O. Aryeetey, Heewon L. Gray, Amos K. Laar, Richard M. K. Adanu.

**Project administration:** Deda Ogum-Alangea, Richmond N. O. Aryeetey, Amos K. Laar.

**Resources:** Richmond N. O. Aryeetey, Richard M. K. Adanu.

**Software:** Heewon L. Gray.

**Supervision:** Deda Ogum-Alangea, Richmond N. O. Aryeetey, Heewon L. Gray, Amos K. Laar, Richard M. K. Adanu.

**Validation:** Richmond N. O. Aryeetey, Heewon L. Gray, Amos K. Laar, Richard M. K. Adanu.

**Visualization:** Deda Ogum-Alangea, Richard M. K. Adanu.

**Writing – original draft:** Deda Ogum-Alangea.

**Writing – review & editing:** Deda Ogum-Alangea, Richmond N. O. Aryeetey, Heewon L. Gray, Amos K. Laar, Richard M. K. Adanu.

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
