## [Decision Letter · Decision Letter 0]

21 May 2020

PONE-D-19-35805

Basic school pupils’ food purchases during mid-morning break in urban Ghanaian schools

PLOS ONE

Dear Dr Ogum Alangea,

Thank you for submitting your manuscript to PLOS ONE. After careful consideration, we feel that it has merit but does not fully meet PLOS ONE’s publication criteria as it currently stands. Therefore, we invite you to submit a revised version of the manuscript that addresses the points raised during the review process.

We would appreciate receiving your revised manuscript by Jul 05 2020 11:59PM. To enhance the reproducibility of your results, we recommend that if applicable you deposit your laboratory protocols in protocols.io, where a protocol can be assigned its own identifier (DOI) such that it can be cited independently in the future. For instructions see: http://journals.plos.org/plosone/s/submission-guidelines#loc-laboratory-protocols

We look forward to receiving your revised manuscript.

Kind regards,

Yacob Zereyesus, Ph.D.

Academic Editor

PLOS ONE

Journal Requirements:

Additional Editor Comments (if provided):

Please see comments from reviewers.

Reviewers' comments:

Reviewer's Responses to Questions

**Comments to the Author**

1. Is the manuscript technically sound, and do the data support the conclusions?

Reviewer #1: Partly

Reviewer #2: Yes

2. Has the statistical analysis been performed appropriately and rigorously? 

Reviewer #1: Yes

Reviewer #2: Yes

3. Have the authors made all data underlying the findings in their manuscript fully available?

Reviewer #1: Yes

Reviewer #2: Yes

4. Is the manuscript presented in an intelligible fashion and written in standard English?

Reviewer #1: Yes

Reviewer #2: Yes

5. Review Comments to the Author

Reviewer #1: This paper examines food purchases within and outside a sample (n=5) of primary and junior secondary schools in Ghana. It shows that children have easy access to energy-dense, high fat and sugar foods, which may contribute to the nutritional transition in this population.

Overall, the paper is written well and it adds to a literature that is limited. However, I feel that it could be strengthened in a number of ways before being published.

1. First, the paper documents what children bought, and by type of vendor within or outside school, but it does not really dig deeper in understanding why children made such choices. In the discussions, it mentions convenience and prices as an important driver of food choice, but there could be others such as taste, emulation of peers, status, etc. Do the authors have collected data on those drivers of food choices among this population? It could be really interesting to understand that more. If they have also qualitative data from focus groups to triangulate with the quantitative findings, the paper could be definitely strengthened, especially for formulating more effective policies on food environments and on nutritional choices in Ghana.

2. The paper would be strengthened by being more specific on its exact contribution to the literature. Especially in the discussions, it mentions many studies from high-income countries, but these have very different food environments and socio-economic drivers of food choices from Ghana. It would be more interesting to compare findings with the literature from low- and middle-income countries, and Sub-Saharan Africa especially.

3. Similarly, in the introduction the authors should mention why we should care about this specific issue, which is prevention of overweight/obesity and related non-communicable disease. At the moment, this point is lacking.

4. The data from the study are relatively old (2013). Based on the speed of the nutrition transition in low- and middle-income countries, we could expect that purchases of unhealthy foods could be even larger now than when these data were collected. Can the authors discuss this point more?

5. Can they clarify if the government schools are part of the Ghana school feeding program? They mention they do not have a school canteen but it is not clear if such program is implemented there. We know that this program could change dietary behaviours of children. Also, many private schools in Ghana also offer school lunches. The authors should explore the aspect of school lunches further and how this relate to food choices among this population, including these considerations in their discussions as well.

6. The paper mentions that it cannot examine drivers of food choices among younger children (up to grade 3) as a limitation. Another limitation however is not examining food choices among older adolescents, as those are even more autonomous than the age groups under consideration in terms of their food choices, and may even more exposed to marketing. Also adolescence is an age in which overweight and obesity start to become manifest among children growing up in low- and middle-income countries (see Schott et al 2019, Economics and Human Biology).

7. In terms of policies, the authors focus on food environments, however nutritional education could also hold promise if combined by shifts towards healthier food environments. Could they discuss this point?

8. The authors should discuss why they selected the specific study settings. Also, how informative are these results, e.g. compared to other major cities in Ghana? And do they have any data on rural areas they could compare their results to?

Minor comments

Line 146: stores, kiosks, etc… why do they start with capital letters?

Line 148: Snack with capital letter

Line 215: variety of food types – “of” missing

Table 2 caption: something missing at the end (I guess “by within/outside school”)

Reviewer #2: Manuscript title: Basic pupils’ food purchases during mid-morning break in urban Ghanaian schools

Manuscript number: PONE-D-19-35805

Reviewer comments

Dear editor,

Thank you for the opportunity to review this manuscript. Below are my comments

General comments

The manuscript is timely for lower middle income settings giving the upsurge in obesity among children and their associated health implications. Since children spend majority of the day when school is in session, they also make decisions regarding what they should eat etc. This decision is usually influence by various factors including meals that are made available within the school and a certain parameter area around the school, what other friends are eating and how much money is giving to the child for snack and food.

My general comment is that this work will benefit from important context information about schools in Ghana, its organization and in particular the operations of school canteens, vendors within and outside of the school compound.

Specific comments

Abstract

1. Background: The first sentence could be modified to: Unhealthy food vending can expose children to malnutrition and other diet related challenges such as obesity. The study sought to…

2. Again to me unhealthy vending resonate more to the vending environment, hygiene practices etc. than the food and its content (energy density) which the study sought to explore.

3. In the methods section of the abstract, you need to indicate how the retail food environment score was estimated. The results on the score had no connection to the content in the methods section

Manuscript

1. Introduction:

a. The authors should include a special section on Ghana, particularly the basic school environment for public and private schools. This will give useful background for the classification of schools under the methods section.

b. Page 5: pages 115-116. Usually the structure of an introduction requires that by the time one is at the conclusion, the issues will be in relation to the study setting (in this case Ghana). Presenting a sentence with a New Zealand example, takes the argument out of place. I suggest New Zealand is not mentioned or that sentence is structured in a way that it relates to Ghana.

2. Methods

a. This section was clearly described. However it is important to explain what table top vendors mean. This may not be a generic/global terminology (page 6, line 144).

3. Results

a. The age range of pupils-I think 21 is an outlier. If this was used in the calculation of the mean age, it may fail to represent the true mean age. If 21 appeared only once, then I suggest the authors can remove that from the analysis. Unless there were many pupils who were around 21 years.

4. Discussion

a. This discussion section has been well-written. I only find some of the comparisons with other studies may not be relevant. For example Page 15, lines 336-337, the North America example can be presented differently

Thank you

6. PLOS authors have the option to publish the peer review history of their article (what does this mean?). If published, this will include your full peer review and any attached files.

Reviewer #1: No

Reviewer #2: No

---

## [Author Response · Author response to Decision Letter 0]

19 Jul 2020

The authors are very grateful to the reviewers of this article. We believe that their comments and suggestions have been very useful in improving the standard of this article. 

Thank you.

---

## [Decision Letter · Decision Letter 1]

14 Aug 2020

Basic school pupils’ food purchases during mid-morning break in urban Ghanaian schools

PONE-D-19-35805R1

Dear Dr. Ogum Alangea,

We’re pleased to inform you that your manuscript has been judged scientifically suitable for publication and will be formally accepted for publication once it meets all outstanding technical requirements.

Kind regards,

Yacob Zereyesus, Ph.D.

Academic Editor

PLOS ONE

Additional Editor Comments (optional):

Reviewers' comments:

Reviewer's Responses to Questions

**Comments to the Author**

1. If the authors have adequately addressed your comments raised in a previous round of review and you feel that this manuscript is now acceptable for publication, you may indicate that here to bypass the “Comments to the Author” section, enter your conflict of interest statement in the “Confidential to Editor” section, and submit your "Accept" recommendation.

Reviewer #1: All comments have been addressed

Reviewer #2: All comments have been addressed

2. Is the manuscript technically sound, and do the data support the conclusions?

Reviewer #1: Yes

Reviewer #2: Yes

3. Has the statistical analysis been performed appropriately and rigorously? 

Reviewer #1: Yes

Reviewer #2: Yes

4. Have the authors made all data underlying the findings in their manuscript fully available?

Reviewer #1: Yes

Reviewer #2: Yes

5. Is the manuscript presented in an intelligible fashion and written in standard English?

Reviewer #1: Yes

Reviewer #2: Yes

6. Review Comments to the Author

Reviewer #1: The authors have satisfactorily addressed my comments throghouly. I am happy for this manuscript to be published.

Reviewer #2: (No Response)

7. PLOS authors have the option to publish the peer review history of their article (what does this mean?). If published, this will include your full peer review and any attached files.

Reviewer #1: No

Reviewer #2: No

---

## [Editor Report · Acceptance letter]

21 Aug 2020

PONE-D-19-35805R1 

Basic school pupils’ food purchases during mid-morning break in urban Ghanaian schools 

Dear Dr. Ogum Alangea:

I'm pleased to inform you that your manuscript has been deemed suitable for publication in PLOS ONE. Congratulations! Your manuscript is now with our production department. 

Kind regards, 

on behalf of

Dr. Yacob Zereyesus 

Academic Editor

PLOS ONE